# Development and Validation of Oral Health-Related Quality of Life Scale for Patients Undergoing Endodontic Treatment (OHQE) for Irreversible Pulpitis

**DOI:** 10.3390/healthcare11212859

**Published:** 2023-10-30

**Authors:** Fadil Abdillah Arifin, Yuhei Matsuda, Takahiro Kanno

**Affiliations:** 1Department of Oral and Maxillofacial Surgery, Shimane University Faculty of Medicine, Izumo 693-8501, Japan; fadilabdillaharifin@umi.ac.id (F.A.A.); yuhei@med.shimane-u.ac.jp (Y.M.); 2Department of Conservative Dentistry, Faculty of Dentistry, Universitas Muslim Indonesia, Makassar 90132, Indonesia

**Keywords:** validation, reliability, quality of life, irreversible pulpitis, endodontic treatment

## Abstract

An oral health-related quality of life measure specific to patients undergoing endodontic treatment has not been developed. This study aimed to validate the oral health-related quality of life scale for patients undergoing endodontic treatment (OHQE) for irreversible pulpitis, comprised of 42 questions. Sixty-two patients with irreversible pulpitis, comprising 23 (37.1%) males and 39 (62.9%) females, were enrolled between August 2022 and February 2023. Data were collected at three time points: pretreatment, post-treatment, and at the second week post-treatment. Factor analysis revealed physical, psychological, and expectations as subscales of OHQE. Cronbach’s alpha coefficients ranged from 0.87 to 0.95 for each subscale. Each subscale of the General Oral Health Assessment Index (GOHAI) was moderately correlated with the OHQE subscales. Good–poor analysis revealed a significant difference between the high-scoring and low-scoring groups for each OHQE subscale. The intraclass correlation coefficients of the OHQE subscales ranged from 0.89 to 0.95. Multivariate linear regression analysis revealed a significant correlation between the pretreatment and post-treatment psychological factors (*p* < 0.05). Thus, OHQE will help researchers and policymakers understand the impact of oral health on the quality of life of patients with irreversible pulpitis undergoing endodontic treatment. OHQE could contribute to the appropriate planning, treatment decisions, and management of dental treatment.

## 1. Introduction

Dental caries affects more than 2.3 billion individuals worldwide and is one of the most common diseases in dentistry [1]. According to the Indonesian Basic Health Research conducted in 2018, the prevalence of caries in Indonesia was 88% [2]. Dental caries is considered a public health problem in countries with weak caries prevention systems. In Mexico, dental caries was detected in 99% of the population [3]. The etiology of dental caries is complex, and it can be caused by a combination of social, psychological, and physical factors. Dental caries is frequently caused by oral micro-organisms, mainly streptococci and lactobacilli, which ferment simple carbohydrates, such as sucrose. The initial indicators of dental caries include surface roughness and subsurface demineralization, which are followed by cavitation, pulp involvement, swelling, abscess formation, and the development of systemic signs and symptoms [4].

Dental caries is the main cause of tooth extraction in patients under the age of 35 [3]. Tooth loss due to dental caries is not limited to oral problems; it is also associated with systemic disease and mortality. A cohort study conducted in China revealed that tooth loss significantly increased the risk of overall death and death from upper gastrointestinal cancer, heart disease, and stroke [5]. Another 12-year cohort study revealed an association between tooth loss and cardiovascular disease [6].

Tooth extraction without prosthodontic treatment affects the entire body, thereby reducing the quality of life (QoL) [7]. Although extraction is more cost-effective than preserving the tooth with endodontic treatment in the short term, the potential need for future replacement of the extracted tooth with an implant, fixed prosthesis, or removable partial dentures indicates that endodontic treatment may be more favorable in terms of cost-effectiveness [8]. More detailed cost-effectiveness calculations for endodontic treatment are expected to be validated using the responsive QoL scale.

The dental pulp comprises a complex arrangement of connective tissue, neurovascular tissue, and humoral cells [9]. Reversible pulpitis elicits spontaneous pain that is relieved a few seconds after the removal of the dental stimulus [10]. Worsening of inflammation and other symptoms, such as discomfort, indicates progression to irreversible symptomatic pulpitis [10,11]. Discomfort can occur suddenly and frequently, followed by heat and sweet sensitivity that persist for a long duration. Dental pain spreads as diffuse pain to the perioral region, thereby reducing the patients’ QoL [11]. Pulp necrosis, the final stage of dental caries, is the collapse of the pulpal defense system against external stimuli, resulting in irreversible damage [9]. Irreversible pulpitis and pulp necrosis require endodontic treatment and can result in the extension of the lesion beyond the apex of the tooth, leading to periapical disease [12,13].

Endodontic treatment involves gaining access to the pulp chamber, chemo-mechanical preparation, and obturation of the root canals [14]. The ultimate goal of endodontic treatment is to render the root canal system bacteria-free and prevent the reinvasion of bacteria and their byproducts from the root canal system into the periradicular tissues [15]. Endodontic treatment is becoming increasingly common owing to patients’ desire to retain their natural teeth and their growing understanding of the advantages of retaining natural teeth. Untreated dental caries can impact the patients’ QoL and psychosocial environment in addition to the mastication function, speech, facial expressions, and psychosocial environment [16]. Wigsten et al. revealed that endodontic treatment improved oral health-related QoL (OHRQoL) more than tooth extraction one month following treatment [17]. OHRQoL is a multidimensional concept that describes the influence of the status of the oral cavity on the function, perceptions, and psychosocial well-being of an individual [18,19]. OHRQoL is an integral part of general health and well-being and is recognized by the World Health Organization as an important segment of the global oral health program [20]. Tools developed and validated for a specific population’s age, local language, and diseases are required to accurately assess the OHRQoL. Several questionnaires have been developed in various languages, including the Geriatric Oral Health Assessment Index (GOHAI) and Oral Health Impact Profile (OHIP) [18].

OHRQoL is measured using GOHAI and OHIP, two widely used questionnaires [18]. GOHAI was originally developed for use in older adult populations [21]; however, it has been used in younger adult populations as a general oral health assessment index in recent years [22]. GOHAI assesses an individual’s perception of their oral health through 12 questions that determine the presence of pain, discomfort, dysfunction, and psychosocial effects of dental diseases [23]. This scale is quick and simple to use and can be self-administered [18]. OHIP was originally developed by Slade and Spencer [24]. A version of OHIP comprising 49 items (OHIP-49) was formulated in Australia based on the statements obtained through interviews with dental patients. These items were distributed across the following seven dimensions elaborated from the theoretical model proposed by Locker: functional limitation, physical pain, psychological discomfort, physical disability, psychological disability, social disability, and handicap [25]. Shortened versions of this instrument have been developed, such as the Oral Health Impact Profile-14 (OHIP-14) [26]. Physical function, pain, and discomfort, which are direct and frequent effects of oral diseases, are considered more important by GOHAI. In contrast, OHIP-14 focuses on the psychological and social impairments [27]. Six of the twelve items in GOHAI focus on functional restrictions or pain and discomfort, whereas ten of the fourteen items in OHIP-14 focus on psychological and behavioral outcomes [22]. Thus, it may be more beneficial to use GOHAI, rather than OHIP-14, in clinical examinations and longitudinal research for each individual, owing to its sensitivity for detecting changes in masticatory performance [28].

GOHAI and OHIP have certain limitations when used for the detection of irreversible pulpitis. The questions on both scales are suitable for assessing general oral health. In contrast, the OHQE scale was developed to assess the physical and psychological QoL of patients with dental pain due to irreversible pulpitis. In addition, these self-reported questionnaires do not report the expectations of endodontic treatment. Thus, this study aimed to develop and validate the oral health-related quality of life scale for patients undergoing endodontic treatment (OHQE) for irreversible pulpitis. The development of the OHQE would aid researchers and policymakers in understanding the impact of irreversible pulpitis and clarifying the decision-making process for endodontic treatment in clinical practice.

## 2. Materials and Methods

### 2.1. Preparation of the Question Pool

The database of PubMed was searched to retrieve relevant publications using MeSH terms such as “irreversible pulpitis”, “endodontic”, and “quality of life”. The retrieved articles were reviewed subsequently. Focus groups with dentists, dental hygienists, and patients were conducted to formulate 89 questions on irreversible pulpitis, endodontics, and QoL. Experts conducted a screening procedure to identify 42 OHQE items suitable for this study by excluding questions with similar connotations or double meanings. The questions were evaluated using a five-point Likert scale, with each score indicating the following: 1, “never”; 2, “rarely”; 3, “sometimes”; 4, “often”; and 5, “very often”. Each subscale of OHQE was rated on a scale of 1–5. Higher ratings indicated a poorer QoL related to oral health in patients with irreversible pulpitis. Thirty-seven OHQE items were selected via the subsequent procedure (Figure 1).

### 2.2. Participants

A total of 359 patients were referred to the Hasanuddin University Dental Hospital in Makassar, Indonesia, for endodontic treatment between August 2022 and February 2023. The inclusion criteria were as follows: (1) age >20 years and (2) dental caries of C3 that reached the dental pulp or higher (ICDAS caries classification). Patients with psychological disorders were excluded from the study [29]. A total of 199 patients were included in this study. All patients received compensation for undergoing endodontic treatment from the funds allocated by the Indonesian government’s National Health Insurance Program. Sixty-eight patients were excluded as they discontinued treatment before completing endodontic treatment. Thus, 62 patients diagnosed with irreversible pulpitis who had completed endodontic treatment, including the follow-up, were included in this study (Figure 2).

Three surveys were conducted to examine the validity and reliability of the new scale. Survey 1 was conducted before commencing endodontic treatment to assess the construct validity, examine the internal consistency, and evaluate the relationship between OHQE and GOHAI. Survey 2 was conducted after the completion of endodontic treatment, including root canal obturation and temporary restoration, to evaluate its predictive validity. A third survey, Survey 3, was conducted at the 2-week follow-up visit. Surveys 2 and 3 were used to assess the test-retest reliability. All patients who had completed the endodontic treatment were instructed to revisit the dental hospital for an evaluation of the results of the endodontic treatment.

### 2.3. Data Collection

#### 2.3.1. Background Data

Data regarding the age (years), sex (male/female), weight (kg), height (m), body mass index (kg/m^2^), employment status (yes/no), university graduation (yes/no), and monthly income in Indonesia’s Rupiah currency (categorized according to the Makassar City regional minimum wage: 3,300,000 IDR ≈ 210 USD), and the number of household members of the patients were collected as background data.

#### 2.3.2. Medical History

Data regarding the medical history of the patients, such as the presence of any systemic disease, medication use, and allergies, was collected. Systemic diseases include diabetes mellitus, hypertension, liver disease, pulmonary disease, thyroid disease, and cancer. In addition, information regarding the medications administered, such as antibiotics, anticancer drugs, calcium channel blockers, and antithyroid drugs, was also collected. The history of any allergies to drugs or food was also recorded.

#### 2.3.3. Dental History

As the dental history of each patient was required in this study, each patient underwent a dental examination to assess the number of teeth, denture use, caries grade, periodontal disease grade, daily brushing frequency, and visual analog scale (VAS) score. Caries grade was categorized into four stages based on the International Caries Detection and Assessment System (ICDAS) in this study: C3 (caries involving the dental pulp) and C4 (the root of the tooth remains) [29]. A new classification from the World Workshop on the Classification of Periodontal and Peri-implant Diseases and Conditions was used to determine the grade of periodontal disease. The grades were as follows: S1, initial; S2, moderate; S3, severe with potential for tooth loss; and S4, severe with potential for loss of all teeth [30].

#### 2.3.4. Questionnaires

The GOHAI and OHQE questionnaires were administered to all participants, which comprised 12 and 42 questions, respectively, following data collection. Both scales were administered three times: before endodontic treatment, after endodontic treatment, and two weeks later after the treatment.

### 2.4. Study Design

This study had a prospective cohort design based on classical test theory, which primarily enables the examination of the validity and reliability of measures of constructs.

### 2.5. Statistical Analysis

Descriptive statistics were used to analyze patient characteristics, with frequency, percentage, median, and standard deviation values according to the variables and their distribution. The Shapiro–Wilk test was used to determine the normality of the data distribution. Five items were removed from the questionnaire item list after item reduction using floor and ceiling effects, which were calculated using the mean value and standard deviation. Three fixed factors were used in the factor analysis with promax rotation to evaluate the construct validity. Cronbach’s alpha coefficients were used to evaluate the internal consistency of the new scale, with each value indicating the following: <0.60, poor; 0.60–0.70, moderate; between 0.70 and 0.80, good; 0.80–0.90, very good; and >0.90, excellent [31]. Spearman’s rank correlation coefficients between each factor of OHQE and GOHAI were calculated to assess concurrent validity. The correlation coefficient interpretations were as follows: 0–0.10, negligible correlation; 0.10–0.39, weak correlation; 0.40–0.69, moderate correlation; 0.70–0.89, strong correlation; and 0.90–1.00, very strong correlation [32]. The good-poor analysis was used to assess discriminant validity, with the median value as a cut-off score for categorical variables. The test-retest reliability of Surveys 2 and 3 was evaluated using the intraclass correlation coefficient (ICC). Each value indicated the following: 0–0.39, poor agreement; 0.40–0.74, modest agreement; and 0.75–1.00, excellent agreement [33]. Multivariate linear regression analysis with forced entry was used to evaluate the predictive validity. Statistical analyses were performed using SPSS (version 27; SPSS Japan Inc., Tokyo, Japan). Two-tailed *p*-values were obtained for all analyses. The alpha level of significance was set at *p* <0.05.

## 3. Results

### 3.1. Participant Characteristic

Among the 199 patients who completed Survey 1, 68 patients dropped out without completing the multivisit treatment. The remaining 131 patients, comprising 62 patients with irreversible pulpitis and 69 patients with pulp necrosis, completed the whole phase of the treatment. The data of patients with irreversible pulpitis were used to validate the new OHQE scale. Among these, 39 (62.9%) were female patients, and 23 (37.1%) were male patients. The mean age of the patients was 36.5 years. Fifty-five patients had C3 caries (88.7%), and seven patients had C4 caries (11.3%). The mean (standard deviation) VAS score for pain was 4.5 (2.7) (Table 1).

### 3.2. Construct Validity and Internal Consistency

Factor analysis revealed three factors (Table 2): physical, psychological, and expectation factors. The Cronbach’s alpha coefficient for the physical, psychological, and expectation factors were 0.95, 0.87, and 0.87, respectively. The cumulative variance of factor loading was 55.9%.

### 3.3. Concurrent Validity

The relationship between the OHQE and GOHAI factors was evaluated for concurrent validity using Spearman’s correlation coefficients. Most of the GOHAI factors were moderately correlated with the physical, psychological, and expectations factors, as well as the total OHQE score. A weak correlation was observed between the psychological component of OHQE and the physical function and pain or discomfort in the mouth item on GOHAI (Table 3).

### 3.4. Good–Poor Analysis for Discriminant Validity

In the high versus low analysis (Figure 3), the Mann–Whitney U test was used to compare the high score group with the low score group for each OHQE factor, which was significantly different between the groups for each factor (*p* < 0.05).

### 3.5. Reliability

The ICC values of OHQE for the physical, psychological, and expectation factors were 0.95, 0.92, and 0.89, respectively. The overall ICC value was 0.58. Except for the total score, which indicates insufficient reliability, all OHQE factors had excellent agreement or sufficient reliability (Table 4).

### 3.6. Predictive Validity

Multivariate linear regression analysis was used to evaluate the predictive validity. The *p*-values for the physical factors, expectation factors, and total score were 0.77, 0.11, and 0.21, respectively. The *p*-value for psychological factors showed a significant correlation (*p* < 0.05; Table 5).

## 4. Discussion

The OHRQoL scales have been used more often for research purposes in dentistry than in clinical practice. GOHAI and OHIP have been applied to patients with various oral diseases owing to the lack of disease-specific OHRQoL scales [34]. GOHAI was originally developed for use in epidemiological studies to evaluate QoL [21]. Therefore, a questionnaire was designed to comprehensively assess the impact of the oral cavity on systemic health [19]. OHQE is a disease-specific scale that can measure the effects of even minor invasive treatments for dental caries. However, several disease-specific measures have been developed for clinical use and epidemiological studies in oncology, an advanced field related to QoL, to assess health status over time when patients are examined using smartphones, as represented by ePRO (electronic patient-reported outcomes (ePROs)) [35,36]. Daily changes in patients with cancer are then used to calculate the clinical minimum important differences (MCID) for research purposes in advance, which is useful for many treatments and supportive care [37,38]. Pain is the main physical manifestation of irreversible pulpitis; however, the treatment strategy must be modified owing to multifactorial influences. Some patients do not seek treatment owing to psychological aspects and progress to pulpal necrosis, whereas others do not have adequate access to dental care due to social aspects (patients for whom extraction is the only option) [39]. Therefore, although OHQE was developed to evaluate the OHRQoL of patients for endodontic research purposes, it is hoped that a shortened version could be developed in the future and that it could be used to calculate MCID to aid in decision-making [40].

Endodontic treatment of irreversible pulpitis and pulp necrosis requires multiple and long-term treatments. Although prolonged endodontic treatment may generate a shift in response to patients undergoing dental treatment, adequate research is lacking. A response shift is a phenomenon specific to patient-reported outcomes (PRO), and its incidence has been reported in dentistry [41]. When individuals self-evaluate their health status, they refer to their internal criteria to make judgments, and the phenomenon of a change in these internal criteria is called a response shift [42]. Response shifts can be classified into three categories: recalibration, reprioritization, and reconceptualization. Recalibration occurs in patients undergoing long-term treatment, especially in patients undergoing endodontic treatment. Recalibration is expected to occur more frequently over the course of long-term treatment in patients undergoing endodontic treatment [43].

The effects of endodontic diseases in terms of clinical characteristics, microbiological factors, and radiographic characteristics are well known from the physicians’ perspective. However, the lack of data on the effects of endodontic diseases from the patient’s perspective is a serious gap in endodontic research. The understanding of oral health from clinicians’ perspectives alone is known to be quite limited. Aside from reporting symptoms, such as pain, endodontists rarely take into account the opinions of their patients. It can be argued that it is important to assess a patient’s QoL, which considers how signs and symptoms affect the patient physically, socially, and psychologically, to determine treatment needs and, ultimately, treatment success [44]. Determining how a certain oral problem affects psychological health or whether it causes patient suffering is another crucial consideration. Oral mucosal disorders, orofacial pain, and tooth loss have all been linked to decreased psychological well-being; however, endodontic status has received less attention [44]. Thus, OHQE could be used to improve the doctor-patient relationship.

Several studies have shown that endodontic treatment can influence the QoL of patients. A prospective longitudinal study reported a significant improvement in OHRQoL after orthograde endodontic treatment. The magnitude of the statistical change was moderate in the short term (one month) and large in the longer term (six months) [45]. A systematic review revealed an increased and widespread interest in the impact of endodontic treatment on QoL; nevertheless, these results are limited to patients who seek endodontic treatment and cannot be generalized [46]. In a previous study, participants from two cities in Canada reported that preoperative factors (e.g., pain and sleep disturbances) affected their QoL, which improved after endodontic treatment. Patient satisfaction improved significantly when endodontic treatment was provided by endodontists [47]. Physical pain was found to be the most affected dimension of the OHRQoL among patients after endodontic treatment at three hospitals in Jeddah, Saudi Arabia [48]. However, given the sensitivity of the QoL scale used in the aforementioned study, the more sensitive OHQE could be used to detect unmet medical needs.

OHQE consists of three subscales: physical, psychological, and expectations. The interpretation of the names of these factors was considered reasonable for the following reasons. In the present study, the pain or discomfort in the mouth component of GOHAI was added to the physical component. OHQE also included expectations as a new subscale, as the perception of patients’ oral health is not accurately reflected in objective assessments of dental problems [19]. The OHRQoL scales are intriguing instruments that assess oral health from the patients’ perspective, assess the patients’ condition or record changes in oral status over the course of treatment, and incorporate the patients’ perceptions and expectations [19,49]. Patients’ expectation fulfillment, adherence, and satisfaction are closely interrelated, which also affect OHRQoL [50]. Patients become dissatisfied if their expectations are not met, and this situation arises when the clinicians’ and patients’ expectations are not aligned. Patients frequently require information; however, this is not always recognized by clinicians, who believe that patients seek prescriptions, tests, or referrals [51].

Internal consistency assesses the degree to which the items on a test are interrelated [52]. The internal consistency was measured using Cronbach’s alpha. Alpha varies from 0 to 1, and high alpha values indicate a high degree of inter-relatedness among the items on a test [52]. Cronbach’s alpha coefficients were calculated as 0.95, 0.87, and 0.87 for the physical, psychological, and expectation factors, respectively, in the present study, indicating that all three factors had excellent internal consistency. Thus, OHQE can be used reliably [31]. These results are similar to those of a study that used the French version of the GOHAI (Cronbach’s alpha = 0.86) [53]. Cronbach’s alpha internal consistency coefficients are influenced by the number of items on the scale [54]. Cronbach’s alpha values were quite low when the number of items was less than 10 [55]. Therefore, the values of the coefficients were expected to be excellent for a scale of over ten items (physical).

The relationship between the OHQE and GOHAI factors was evaluated for concurrent validity using Spearman’s correlation coefficients. Spearman rank correlation describes the monotonic relationship between two variables. This correlation coefficient is (1) useful for non-normally distributed continuous data, (2) applicable to ordinal data, and (3) relatively robust for outliers [56]. Spearman’s coefficient ranges from −1 to +1, which can be interpreted as describing anything between no association (*r* = 0) and a perfect monotonic relationship (*r* = −1 or +1) [56]. Correlation coefficients describe the strength and direction of the association between the variables [56]. The scores of the correlation coefficients in each of the OHQE subscales ranged from 0.18 to 0.61, and most of the relationship between OHQE and GOHAI produced a moderate correlation [32]. Excluding the correlation between the psychological factors of OHQE and the pain or discomfort in the mouth item and the physical function of GOHAI was weak [32]. The results were 0.18 and 0.21, respectively). In addition, all correlation values from each factor were positive, indicating that they were correlated with each factor. The relationship between OHQE and GOHAI produced reasonable values in terms of validity and reliability. Moreover, OHQE showed satisfactory concurrent validity with significant correlations with GOHAI, similar to a study using GOHAI in the Greek language [57]. This establishes the status of concurrent validity as external validity.

Test-retest reliability was assessed by calculating the ICCs [52]. Absolute reliability refers to the degree to which repeated measurements of the same instrument on the same individual vary around the true score. A smaller variation in repeated measurements indicates a higher absolute reliability [52]. In this study, the ICC values for each OHQE factor indicated acceptable reliability. This favorable value is most likely a result of the fact that there was only a two-week interval between Surveys 2 and 3, which were utilized as the control visits for the test-retest reliability assessment. The new scale has 42 items. The reliability of the test increases as the sample of items taken from a given area of knowledge and skill increases. The difficulty level, clarity of expression, and conciseness of instructions for a test item also affect the reliability of the test scores. If the test items are very easy or difficult for group members, it will lead to low-reliability scores. This may be attributed to a restricted spread of scores in both tests. The discriminant validity of OHQE was verified, as statistically significant differences were observed between the low-scoring and high-scoring groups of each subscale (*p* < 0.05) divided by the cut-off score using the median value. The results of predictive validity indicated that psychological factors are likely to play a role in patients with irreversible pulpitis undergoing endodontic treatment.

However, this study has some limitations. The sample size is the first limitation of this study. The sample size was expected, given the dropout rate (10%). However, the number of individuals who withdrew from the study (34.2%) was higher than predicted. Thus, there could be a nonrespondent bias in this study [58]. The patients elected against continuing the endodontic procedure, probably owing to the requirement of multiple appointments and lengthy waiting periods. Endodontic therapy requires a minimum of seven appointments in accordance with Indonesian health insurance regulations. In addition, some patients no longer experienced tooth pain after a few initial visits and elected not to continue the sequential treatment. Therefore, further research with larger sample sizes is necessary, as the study sample did not accurately reflect the community of endodontic patients. Second, patient participation is another drawback. Some patients thought it would be time-consuming for them to read and respond to all the questions. Lastly, the current study only represents the condition of one tooth that requires endodontic treatment, not the complete oral health status of the participants.

The newly developed OHQE has several advantages, including the ability to assess the QoL of patients with irreversible pulpitis who experience physical and psychological symptoms related to dental pain. Furthermore, this new scale can determine the expectations of patients with irreversible pulpitis from endodontic therapy. These benefits can be used by future researchers and policymakers to better understand the influence of oral health on the QoL of patients with irreversible pulpitis and aid in the appropriate planning and management of dental health programs.

## 5. Conclusions

Numerous statistical analyses have verified that OHQE is a reliable and valid scale that can be used to measure the OHRQoL in patients with irreversible pulpitis undergoing endodontic treatment as a disease-specific scale.

## Figures and Tables

**Figure 1 healthcare-11-02859-f001:**
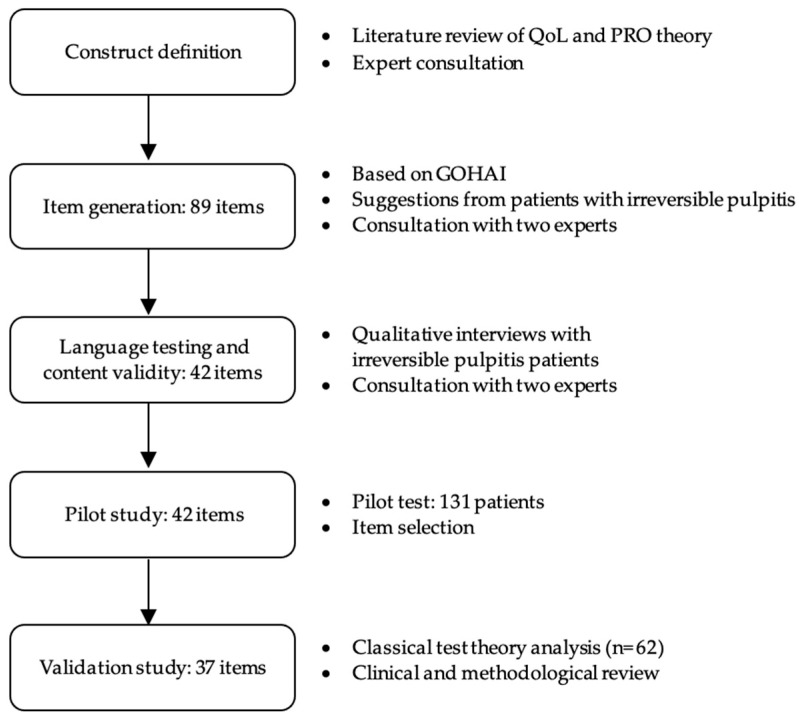
Development of the question pool. QoL, Quality of Life; GOHAI, General Oral Health Assessment Index; PRO, Patient-Reported Outcome.

**Figure 2 healthcare-11-02859-f002:**
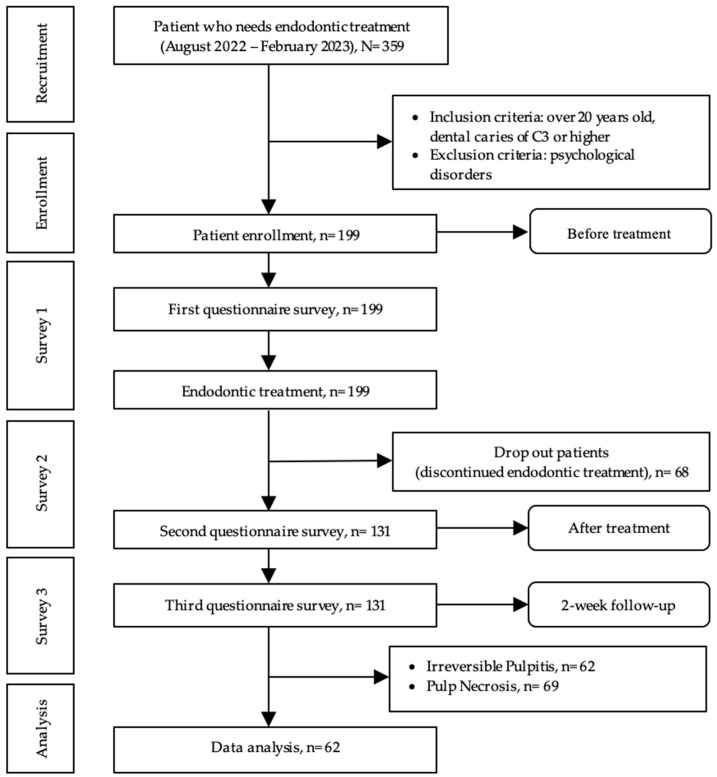
Flow chart outlining the study.

**Figure 3 healthcare-11-02859-f003:**
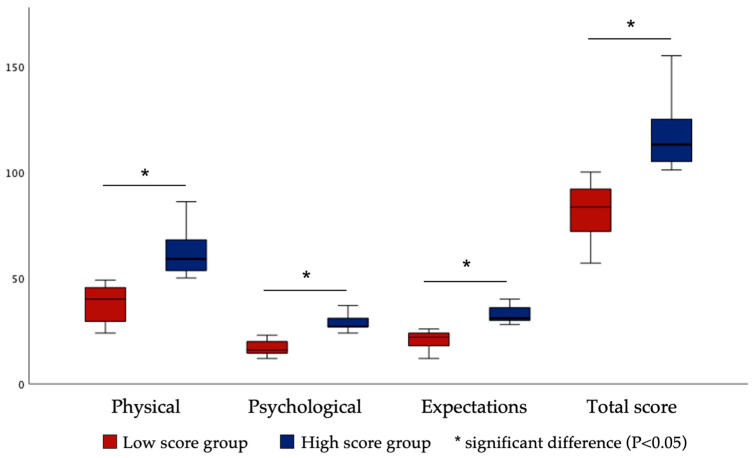
Discriminant validity according to the Mann–Whitney U test.

**Table 1 healthcare-11-02859-t001:** Demographic characteristics of the participants (*n* = 62).

Variables	Categories	*n* (%) or Mean [SD]
Age (years)		36.5 [12.4]
Sex	Male	23 (37.1)
	Female	39 (62.9)
Body mass index (kg/m^2^)		23.5 [3.6]
Job	Employed	34 (54.8)
	Unemployed	28 (45.2)
University graduate	Yes	32 (51.6)
	No	30 (48.4)
Monthly income	<3,300,000 IDR	43 (69.4)
	≥3,300,000 IDR	19 (30.6)
Number of housemates		4.2 [1.6]
Systemic disease	Yes	15 (24.2)
	No	47 (75.8)
Medication taken	Yes	8 (12.9)
	No	54 (87.1)
Allergies	Yes	5 (8.1)
	No	57 (91.9)
Number of teeth		26.7 [4.3]
Denture use	Yes	4 (6.5)
	No	58 (93.5)
Grade of caries	C3	55 (88.7)
	C4	7 (11.3)
Grade of periodontal disease	S1	37 (59.7)
	S2	21 (33.9)
	S3	4 (6.5)
	S4	0 (0.0)
Brushing (times)		2.2 [0.5]
VAS		4.5 [2.7]

VAS, visual analogue scale; SD, standard deviation.

**Table 2 healthcare-11-02859-t002:** Rotated factor loadings for construct validity and internal consistency.

Items	Factor Loading		
	1	2	3
Factor 1. Physical			
Q10. Have you had toothache?	0.81	−0.32	0.17
Q14. Have you ever felt pain that radiates from a dental pain?	0.79	0.34	0.07
Q24. Has your sleep been interrupted because of dental pain?	0.79	0.31	−0.03
Q9. Have you had headaches because of dental pain?	0.76	0.24	−0.08
Q8. Have you had a sore jaw because of dental pain?	0.75	0.19	0.29
Q7. Have you had painful aching in your mouth because of dental pain?	0.71	0.17	0.06
Q11. Have you had painful gums?	0.70	0.05	0.15
Q26. Have you found it difficult to relax because of dental pain?	0.68	0.53	0.14
Q13. Have you ever taken medication to relieve dental pain?	0.67	0.11	0.19
Q15. Have you ever felt pain radiating to the ear because of dental pain?	0.67	0.23	0.26
Q1. Have you had difficulty chewing any foods because of dental pain?	0.65	0.26	−0.002
Q6. Have you ever had difficulty opening your mouth because of dental pain?	0.63	0.53	−0.06
Q23. Have you ever been unable to lie down because of dental pain?	0.62	0.37	0.09
Q21. Have you had to avoid eating some foods because of dental pain?	0.62	0.43	0.11
Q27. Have you avoided going out because of dental pain?	0.59	0.53	−0.17
Q17. Have you felt tense because of dental pain?	0.57	0.54	0.13
Q18. Have you ever felt that your toothache is a serious disease?	0.54	0.44	0.31
Q28. Have you been a bit irritable with other people because of dental pain?	0.45	0.39	−0.15
Q33. Have you ever thought that the root canal treatment can be reinfected in the future?	0.36	0.11	−0.37
Factor 2. Psychological			
Q16. Have you felt uncomfortable about the appearance of your teeth because of dental pain?	0.12	0.84	0.22
Q4. Have you felt that your sense of taste has worsened because of dental pain?	0.11	0.82	0.01
Q5. Have you felt that your digestion has worsened because of dental pain?	0.11	0.78	0.14
Q3. Have you felt that your appearance has been affected because of dental pain?	0.15	0.73	−0.008
Q19. Do you ever overthink about your health condition because of dental pain?	0.41	0.59	0.27
Q2. Have you had trouble pronouncing any words because of dental pain?	0.39	0.55	−0.11
Q29. Have you felt that your general health has worsened because of dental pain?	0.29	0.54	0.16
Q25. Have you been upset because of dental pain?	0.47	0.54	0.21
Q20. Has your speech been unclear because of dental pain?	0.38	0.41	−0.29
Q31. Have you ever thought that it is better to have a tooth extracted than to treat it?	0.17	0.29	−0.03
Factor 3. Expectations			
Q37. Have you ever thought that root canal treatment could have a good impact on your health?	−0.07	0.00	0.82
Q22. Have you ever thought that root canal treatment can improve your chewing function?	0.15	−0.08	0.79
Q35. Have you ever thought that root canal treatment can improve quality of life?	0.03	0.12	0.79
Q36. Have you ever thought that root canal treatment can improve dental aesthetics?	−0.05	0.30	0.77
Q12. Have you ever felt that root canal treatment can eliminate your dental pain?	0.18	0.01	0.76
Q32. Have you ever thought that root canal treatment is worth doing?	0.09	−0.04	0.65
Q30. Have you ever thought that root canal treatment is expensive?	0.12	0.17	0.55
Q34. Have you ever thought that root canal treatment should be performed by a specialist rather than a general dentist?	0.26	0.05	0.51
Sum of squares on factor loading	9.29	6.43	4.98
Variance explained (%)	25.13	17.38	13.45
Cumulative variance explained (%)	25.13	42.50	55.95
Cronbach’s alpha coefficient	0.95	0.87	0.87

**Table 3 healthcare-11-02859-t003:** Relationship between OHQE and GOHAI for concurrent validity.

GOHAI	OHQE			
	Physical	Psychological	Expectations	Total Score
Physical function				
r	0.28	0.21	0.36	0.38
*p*-value	0.03 *	0.09	0.004 *	0.003 *
Psychosocial function				
r	0.48	0.57	0.37	0.61
*p*-value	<0.01 *	<0.01 *	0.003 *	<0.01 *
Pain or discomfort in the mouth				
r	0.27	0.18	0.29	0.31
*p*-value	0.03 *	0.15	0.02 *	0.02 *
Total score				
r	0.44	0.45	0.41	0.57
*p*-value	<0.01 *	<0.01 *	0.001 *	<0.01 *

OHQE: oral health-related quality of life scale for patients with endodontic disease; GOHAI: general oral health assessment index; * significant difference (*p* < 0.05).

**Table 4 healthcare-11-02859-t004:** Test-retest reliability of OHQE using ICC.

Factor	ICC (95% CI)
Physical	0.95 (0.93–0.96)
Psychological	0.92 (0.89–0.95)
Expectations	0.89 (0.84–0.93)
Total score	0.58 (0.18–0.77)

ICC: intraclass correlation coefficient, CI: confidence interval.

**Table 5 healthcare-11-02859-t005:** Multivariate analysis using linear regression analysis for predictive validity.

Variables	β	B	95% CI		*p*-Value
			Lower	Upper	
Physical	0.04	0.05	−0.29	0.39	0.77
Psychological	0.26	0.74	0.02	1.45	0.04 *
Expectations	0.21	0.60	−0.13	1.33	0.11
Total score	0.16	0.14	−0.08	0.35	0.21

* significant difference (*p* < 0.05); CI: confidence interval.

## Data Availability

The data presented in this study are available upon request from the corresponding author. These data are not publicly available for ethical reasons.

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
