# Peer review of "Development and Validation of Oral Health-Related Quality of Life Scale for Patients Undergoing Endodontic Treatment (OHQE) for Irreversible Pulpitis"

_healthcare, 2023, doi:10.3390/healthcare11212859_

Round 1
Reviewer 1 Report
Comments and Suggestions for Authors
Dear Authors.
After reviewing the document, I have to make the following comments and suggestions:
The abstract seems to me adequate, although its different sections should be specified as indicated by the journal's standards: Background, Methods, Results and Conclusions.
The background of this article is well elaborated describing extensively the existing problems on the subject of research because, although there are scales to evaluate oral health in general, there is no scale that measures the physical and psychological quality of life related to oral health in patients with dental pain due to irreversible pulpitis who are going to undergo endodontic treatment.
The objective is adequate to clarify the study problem posed.
The methodology allows the authors to adequately address the study problem to achieve the proposed objective, although:
- The causes for the elimination of items in the pilot study should be specified (Figure 1).
- The methodology used should be described, even briefly, and not refer to a reference (Lines 201-202).
- The Shapiro-Wilk test is not indicated since there are more than 50 subjects in the study, the Kolmogorov-Smirnov test being more appropriate (Lines 206-207).
- To analyze internal consistency, the McDonald omega test would be more appropriate since it is an ordinal variable (Likert scale). However, if the distribution was not normal, the ordinal alpha test would be more appropriate (Lines 210-211).
The results have been presented in a clear way to facilitate understanding, although:
- It would be more convenient to indicate the mean and standard deviation of quantitative variables and the frequencies and percentages of qualitative variables (Lines 205-206 and Table 1).
- In Table 2 question 31 has a very low factor loading (0.29) with factor 2 (Psychological), wouldn't it have been better to eliminate it from the scale?
The discussion provides a detailed analysis of the results obtained and establishes relationships with previous studies on the subject of this research. Although from my point of view there is excessive emphasis on the pathology and on the need to carry out this questionnaire, points that have already been dealt with previously in the introduction.
The conclusion is coherent with the results obtained and responds to the proposed objective.
The references are extensive and appropriate to address this research topic.
Kind regards.
Author Response
Responses to Comments/Suggestions
by Reviewer 1
Dear Reviewer
We are truly grateful for your critical comments and thoughtful suggestions for our manuscript, which have helped us further improve our manuscript. Based on these comments and suggestions, we have carefully revised our original manuscript. All changes made to the main text are presented in red font. Please find our point-by-point responses to your comments/questions below.
Sincerely yours
Prof. Takahiro Kanno,
Corresponding Author
(Manuscript ID: healthcare-2667631)
Reviewer’s comments:
After reviewing the document, I have to make the following comments and suggestions:
The abstract seems to me adequate, although its different sections should be specified as indicated by the journal's standards: Background, Methods, Results and Conclusions.
Response: Thank you very much for this suggestion. As you mentioned, in general, abstracts are often structured. However, the MDPI Guidelines for Authors instruct authors to use a single paragraph when submitting to Healthcare without structuring the abstract. Therefore, no modifications have been made to the abstract.
The background of this article is well elaborated describing extensively the existing problems on the subject of research because, although there are scales to evaluate oral health in general, there is no scale that measures the physical and psychological quality of life related to oral health in patients with dental pain due to irreversible pulpitis who are going to undergo endodontic treatment.
Response: Thank you very much for your comment. We believe that this scale will facilitate the evaluation of patients with irreversible pulpitis patients.
The objective is adequate to clarify the study problem posed.
Response: Thank you for this comment. Efforts were made to clarify the objectives.
The methodology allows the authors to adequately address the study problem to achieve the proposed objective, although:
- The causes for the elimination of items in the pilot study should be specified (Figure 1).
Response: Thank you for this suggestion. When creating an item pool, it is necessary to create as many question items as possible. However, various questions were screened and excluded by experts because of the homogeneity among questions and some unclear questions (e.g., double-barrel questions). We have further clarified that many questions were excluded, as you indicated (Line 124-125).
- The methodology used should be described, even briefly, and not refer to a reference (Lines 201-202).
Response: Thank you for this suggestion. A prospective cohort study design based on classical test theory was selected for this study, which primarily examines the validity and reliability of construct measures. As you suggested, we have added a brief note on that point (Line 200-201).
- The Shapiro-Wilk test is not indicated since there are more than 50 subjects in the study, the Kolmogorov-Smirnov test being more appropriate (Lines 206-207).
Response: Thank you for this suggestion. As you pointed out, the Kolmogorov-Smirnov test could have been used. However, it is still under debate whether the Shapiro-Wilk test or the Kolmogorov-Smirnov test should be used for different sample sizes. The Kolmogorov-Smirnov test is said to be more suitable when using large samples. However, since we consider the sample size in this study to be small, we have not changed our analysis method.
- To analyze internal consistency, the McDonald omega test would be more appropriate since it is an ordinal variable (Likert scale). However, if the distribution was not normal, the ordinal alpha test would be more appropriate (Lines 210-211).
Response: Thank you for this suggestion. As you pointed out, an increasing number of studies have used McDonald's Omega in recent years as a way to account for the influence (factor loadings) on response values to each question from common factors. However, COSMIN guidelines for the development of medical scales still recommend the use of Cronbach's alpha; thus, we have not modified it.
The results have been presented in a clear way to facilitate understanding, although:
- It would be more convenient to indicate the mean and standard deviation of quantitative variables and the frequencies and percentages of qualitative variables (Lines 205-206 and Table 1).
Response: Thank you for this suggestion. We have added the means and standard deviations of quantitative values in the text and Table 1.
- In Table 2 question 31 has a very low factor loading (0.29) with factor 2 (Psychological), wouldn't it have been better to eliminate it from the scale?
Response: Thank you for this question. As you mentioned, statistically, we should consider removing it; however, we believe it is a very important clinical question. In particular, this question is one of the questions that characterize this scale and plays a role in creating its disease-specificity. We did not eliminate question number 31 (Have you ever thought that it is better to have a tooth extracted than to treat it?), because we need patients’ perceptions and preferences regarding tooth extraction or maintenance with an endodontic treatment.
The discussion provides a detailed analysis of the results obtained and establishes relationships with previous studies on the subject of this research. Although from my point of view there is excessive emphasis on the pathology and on the need to carry out this questionnaire, points that have already been dealt with previously in the introduction.
Response: Thank you for these insights. In root canal treatment, the pathology of the pulp is strongly related to the clinical findings; thus many pathological references have been added to the discussion.
The conclusion is coherent with the results obtained and responds to the proposed objective.
Response: Thank you very much.
The references are extensive and appropriate to address this research topic.
Response: Thank you very much.
Reviewer 2 Report
Comments and Suggestions for Authors
Manuscript title: Development and Validation of Oral Health-Related Quality of Life Scale for Patients undergoing Endodontic Treatment (OHQE) for Irreversible Pulpitis requires modifications before considered for publication.
Abstract: Material method and result section requires more explanation. Keywords: simple and understandable.
Introduction: Too lengthy and few lines does not make any sense, for example line 38 and 39, complete paragraph starting from line 56-70. Try to concise introduction. However, the need for the study is explained properly.
Material and method: Explained well but too lengthy, because of which readers might lose interest.
Result: Well-presented and tables and figures are properly marked.
Discussion: Too lengthy and at many places sentences do not make sense.
Conclusion: Try to add a few more points here to support your questionnaire.
References are properly marked, and no duplication is seen.
Comments on the Quality of English LanguageMajor editing required
Author Response
Responses to Comments/Suggestions
by Reviewer 2
Dear Reviewer
We are truly grateful for your critical comments and thoughtful suggestions for our manuscript, which have helped us further improve our manuscript. Based on these comments and suggestions, we have carefully revised our original manuscript. All changes made to the main text are presented in red font. Please find our point-by-point responses to your comments/questions below.
Sincerely yours
Prof. Takahiro Kanno,
Corresponding Author
(Manuscript ID: healthcare-2667631)
Reviewer’s comments:
Manuscript title: Development and Validation of Oral Health-Related Quality of Life Scale for Patients undergoing Endodontic Treatment (OHQE) for Irreversible Pulpitis requires modifications before considered for publication.
Response: Thank you for this suggestion. We have confirmed that the title of the manuscript is suitable and truly representative of the content of the paper.
Abstract: Material method and result section requires more explanation. Keywords: simple and understandable.
Response: Thank you for this suggestion. Keywords were chosen based on MeSH terms whenever possible, so simplified terms were used and have not been changed.
Introduction: Too lengthy and few lines does not make any sense, for example line 38 and 39, complete paragraph starting from line 56-70. Try to concise introduction. However, the need for the study is explained properly.
Response: Thank you for this suggestion. As you indicated, the background may be somewhat extensive. However, the journal does not have a maximum but rather a minimum limit of 4,000 words; thus, we have included a longer description because detailed descriptions in each section are recommended by this journal.
Material and method: Explained well but too lengthy, because of which readers might lose interest.
Response: Thank you for these insights. For the reasons already detailed above, we have not made any changes.
Result: Well-presented and tables and figures are properly marked.
Response: Thank you for this comment.
Discussion: Too lengthy and at many places sentences do not make sense.
Response: Thank you for these insights. For the reasons already detailed above, we have not made any changes.
Conclusion: Try to add a few more points here to support your questionnaire.
Response: Thank you for this suggestion. Accordingly, we have emphasized in the conclusion section that the scale developed in this study is a disease-specific scale.
References are properly marked, and no duplication is seen.
Response: Thank you.
Reviewer 3 Report
Comments and Suggestions for Authors
Good manuscript with proper scientific structure. Clearly stated thesis addressed by analysis and discussion. Comprehensible English. The manuscript introduces potential new scale to evaluate quality of life, of patients undergoing endodontic treatment. Although, questionnaire was investigated only on 359 patients, it demonstrated potential for its usefulness and will contribute to the existing literature. Article suitable for publication if authors address following issues:
1. Ln 124-126. Why did authors select those search terms? What were the inclusion criteria? Please explain
2. Ln 161-164 – This paragraph should be placed at the end of the manuscript, before references. Current position is not appropriate. It should be improved.
3. Ln 171 – please provide this wage in USD. This is the most common currency, and the values are the easiest to understand.
4. Ln 209-2012 – Why did authors use Cronbach’s coefficients? Please explain
5. Ln 428-433 – Conclusions should refer only to authors own analysis and results. Current conclusions are too vague. Please precise and expand this section with 3-4 sentences.
Author Response
Responses to Comments/Suggestions
by Reviewer 3
Dear Reviewer
We are truly grateful for your critical comments and thoughtful suggestions for our manuscript, which have helped us further improve our manuscript. Based on these comments and suggestions, we have carefully revised our original manuscript. All changes made to the main text are presented in red font. Please find our point-by-point responses to your comments/questions below.
Sincerely yours
Prof. Takahiro Kanno,
Corresponding Author
(Manuscript ID: healthcare-2667631)
Reviewer’s comments:
Good manuscript with proper scientific structure. Clearly stated thesis addressed by analysis and discussion. Comprehensible English. The manuscript introduces potential new scale to evaluate quality of life, of patients undergoing endodontic treatment. Although, questionnaire was investigated only on 359 patients, it demonstrated potential for its usefulness and will contribute to the existing literature. Article suitable for publication if authors address following issues:
- Ln 124-126. Why did authors select those search terms? What were the inclusion criteria? Please explain
Response: Thank you for this question. The research plan for this study was based on the COSMIN checklist, While the COSMIN checklist includes a literature review section, there are no specific rules regarding the criteria for selecting search words. Therefore, this study conducted a literature search based on the researcher's narrative.
- Ln 161-164 – This paragraph should be placed at the end of the manuscript, before references. Current position is not appropriate. It should be improved.
Response: Thank you for this suggestion. Accordingly, we have moved the ethics approval section to the end of the manuscript (Line 438-440).
- Ln 171 – please provide this wage in USD. This is the most common currency, and the values are the easiest to understand.
Response: Thank you for this suggestion. We have added a USD value for the wage to the revised manuscript.
- Ln 209-2012 – Why did authors use Cronbach’s coefficients? Please explain
Response: Thank you for this question. As specified in the COSMIN checklist, it is recommended to verify the internal consistency of factor-specific questionnaire items when developing multifactor scales. Cronbach's alpha was used because it is the best method of validating internal consistency.
- Ln 428-433 – Conclusions should refer only to authors own analysis and results. Current conclusions are too vague. Please precise and expand this section with 3-4 sentences.
Response: Thank you for this suggestion. As suggested, we have summarized and specified the conclusions based on our results. (Line 427-429)
Round 2
Reviewer 1 Report
Comments and Suggestions for Authors
Dear Authors.
I consider that the manuscript has been sufficiently improved and I have no additional comments or suggestions.
Kind regards.
Author Response
Dear Reviewer 1,
Thank you very much for your kind comment.
We hope this paper will contribute to dentistry.
Best regards,
All authors
Reviewer 2 Report
Comments and Suggestions for Authors
Dear Authors,
Thank you for your comments.
Author Response
Dear Reviewer 2,
Thank you very much for your kind comment.
We hope this paper will contribute to dentistry.
Best regards,
All authors